# Orthodontic Risk Perspectives among Orthodontists during Treatment: A Descriptive Pilot Study in Greece and Slovakia

**DOI:** 10.3390/healthcare12040492

**Published:** 2024-02-18

**Authors:** Franzeska Karkazi, Maria Antoniadou, Katarína Demeterová, Dimitrios Konstantonis, Vasileios Margaritis, Juraj Lysy

**Affiliations:** 1Department of Orthodontics, School of Health Sciences, Faculty of Dentistry, Marmara University, Istanbul 34722, Turkey; fkarkazi@aol.com; 2Dental School, National and Kapodistrian University of Athens, 11527 Athens, Greece; dikons@dent.uoa.gr; 3Certified Systemic Analyst Executive Mastering Program, University of Piraeus, 18534 Piraeus, Greece; 4Department of Stomatology and Maxillofacial Surgery, Faculty of Medicine, Comenius University Bratislava, 81250 Bratislava, Slovakia; demeterova20@uniba.sk; 5Research Center, Swiss School of Management, 6500 Bellinzona, Switzerland; vasileios.margaritis@ssm.swiss

**Keywords:** orthodontic risk, informed consent, patient-centered care, cross-cultural practices, treatment risks, digital tools, patient satisfaction, ethical considerations

## Abstract

This study explores orthodontists’ perspectives on risks associated with orthodontic treatment, as described by Greek and Slovak orthodontists. Informed by the foundational importance of effective communication of risk perspectives in health sciences, particularly in facilitating valid consent and shared decision-making, this research addresses gaps identified in the literature concerning the consistent communication of potential treatment risks based on demographic and cultural characteristics. This study identifies 15 potential critical risks during orthodontic treatment. These risks include root resorption; temporary undesired changes to the occlusion; sleep difficulties; not achieving an ideal result; development of black triangles between teeth; taking additional X-rays; speech difficulties; using a protective splint during sports; duration of treatment; number of visits; transmission of infectious diseases; and swallowing orthodontic appliances. A questionnaire, distributed electronically to orthodontists in Greece (N1 = 570) and Slovakia (N2 = 210) from September 2022 to December 2022, aimed to assess risk communication practices, taking into consideration socio-demographic factors, such as country, gender, age, and academic-degree-related variations. A total of 168 valid questionnaires (91 from Slovakia and 77 from Greece) were obtained, indicating significant disparities in the risks emphasized and preferred forms of consent. The Greek orthodontists focused more on the risks involved, such as relapse, root resorption, temporal occlusal changes, and failure of desired movement, while the Slovak practitioners tended to be more interested in sleeping difficulties, temporal occlusal changes, and not achieving an ideal result. They also obtained written or digital consent from patients or their parents/guardians more frequently than the Greek team. Male orthodontists discussed specific risks more frequently, including relapse and extractions, whereas females preferred written or digital consent. PhD-trained orthodontists prioritized certain risks, indicating the need for tailored approaches. This study underscores the dynamic nature of risk assessment in orthodontic practice, emphasizing its ethical and strategic dimensions. The findings advocate for tailored risk communication strategies that recognize individual, contextual, and cultural factors, and the need for an orthodontic informed consent protocol for a tailored communication approach for patients to elevate the standard of care in European orthodontics. The reliance on digital tools reflects contemporary trends in enhancing patient understanding, thereby supporting ongoing innovation in orthodontic practices.

## 1. Introduction

In healthcare, the concept of informed consent, first introduced in 1957, stands as a pivotal ethical cornerstone, ensuring the safeguarding of patients’ and research participants’ welfare [1]. This indispensable concept encapsulates a communicative process wherein patients are provided comprehensive information regarding the potential benefits and risks associated with a proposed treatment. In reciprocation, the patient, armed with this knowledge, consents to the recommended course of action or intervention. The validity of informed consent hinges upon several fundamental criteria, each playing a crucial role in upholding ethical standards [2]. These include the patient’s capacity to comprehend the relevant facts about the proposed treatment and available choices, the voluntary nature of the consent, to prevent coercion, and the necessity for the consent to be informed and sufficiently specific [3]. While written consent might not be imperative for straightforward procedures, its significance becomes paramount in the grounds of complex, extensive, or multifaceted interventions, such as those encountered in orthodontics [4]. Orthodontists, like all health care professionals, have an ethical obligation to communicate the potential risks and benefits before starting treatment so that patients are aware and prepared for any adverse conditions or problems that could arise from the treatment [5]. While an orthodontist is guided to diagnosis and treatment planning by objective factors that are derived from clinical screening and adequate analyses, orthodontic patients are driven by subjective factors, like their perception of the problem, their needs, and desires [6,7]. Therefore, to effectively address patient concerns and expectations, it is imperative to adopt a patient-centered approach in orthodontic practice [8,9]. In the contemporary dental landscape, risk assessment emerges not only as a fundamental aspect of professional management but also as a strategic component in the marketing strategy of orthodontic practices. This assessment is seamlessly integrated into the evaluation of any orthodontic anomaly and its recommended treatment, extending its relevance to post-treatment care policies [8].

Thus, an informed consent that effectively bridges the gap between the clinician’s viewpoint and the expectations of patients necessitates the inclusion of comprehensive information. This should encompass details about treatment goals and alternative approaches. Moreover, critical aspects such as patient cooperation, anticipated treatment duration, potential discomfort, and various adverse effects during or after the treatment must be meticulously communicated [8,9]. By providing a thorough overview of these elements, the informed consent process ensures transparency and empowers patients to make informed decisions about their orthodontic care [6].

In the literature, the identified risks associated with orthodontic treatment span a broad spectrum, encompassing periodontal concerns such as gingivitis or bone loss [10,11,12], as well as issues like cavities [13], root resorption [14], and tooth necrosis [15]. Inadequate oral hygiene may lead to the emergence of caries or enamel decalcifications [10,13,16]. Furthermore, patients may experience altered discomfort or pain [10,17], difficulties in speech [18], especially at the onset of treatment, or psychological distress stemming from the visibility of orthodontic appliances [19,20,21]. Allergic reactions to materials used in treatment have also been documented, introducing an additional layer of consideration [22,23]. Aesthetic challenges, including the appearance of black triangles [24,25] and the potential failure of tooth displacement [10], add complexity to orthodontic interventions. Patients must also be informed about potential extraction needs [26,27], surgical procedures, and the likelihood of relapse [10]. Additional risks involve appliance breakage, detached brackets, broken or protruding wires, time and cost for repairs, and even the possibility of inadvertently swallowing orthodontic components [28,29,30]. Notably, the recent focus on infectious disease transmission, particularly in the context of the COVID-19 pandemic, has underscored the need for heightened attention to infection control within dental offices [31]. Finally, Perry et al. (2021) [32] conducted a comprehensive analysis, identifying 30 evidence-based risks in orthodontic treatment, culminating in the identification of 10 critical risks that should be consistently communicated to patients. These risks include demineralization/caries; relapse; length of treatment; root resorption; pain/discomfort; the consequences of doing nothing; appliances breaking; failure to achieve desired tooth movements; gingivitis; and mucosal ulcerations.

Despite the amount of information available, there exist notable gaps in the current body of research. While the literature provides valuable insights into potential risks associated with orthodontic treatment, a comprehensive and standardized approach to risk communication and patient education in different cultures is lacking. The development of clear protocols and effective communication strategies is imperative to enhance patient understanding and engagement throughout the orthodontic treatment process, especially now that patients are even more informed and expect more from their treatment [20,33]. This will not only contribute to the refinement of clinical practices but also empower patients to make well-informed decisions about their orthodontic care [5].

In this sense, the ethical consideration of informed consent not only ensures the welfare of orthodontic patients but is currently an important issue in dentistry [9,34]. This study addresses the gap by reporting on the extent of informed consent practices in orthodontics in two European countries, namely Greece and Slovakia, offering an extensive analysis of orthodontists’ perspectives shaped by individual and cultural factors. The primary objectives of this study were to identify potential critical risks associated with orthodontic treatment used by professionals in the field, address issues related to consistent risk communication, evaluate variations in risk communication practices based on demographic factors, and understand orthodontists’ preferences for obtaining consent. It also aimed to contribute valuable insights that can shape the development of tailored risk communication strategies to enhance the standard of care in European orthodontics. Additionally, the study acknowledged the influence of contemporary trends, discussing the role of digital tools in patient understanding and supporting ongoing innovation and management in orthodontic practices. Overall, using a systematic questionnaire study format, the study aimed to investigate the patterns, prevalence, and severity of orthodontic risks and contribute to updated informed guidelines for risk assessment in orthodontic practices.

## 2. Materials and Methods

### 2.1. Designing the Study Questionnaire

Participants in this study were professional orthodontists in Slovakia (N1 = 210) and Greece (N2 = 570). Non-orthodontists, dentists of other specialties, and undergraduate and postgraduate dental students were excluded. 

The questionnaire design was executed in two rounds. Firstly, it underwent preliminary testing conducted in September 2022. This initial phase involved a small sample of orthodontists from the Dental School of Athens, Greece. The primary objective was to assess the feasibility and clarity of the preliminary questionnaire. The questionnaire was initially designed in English and reviewed by an English-speaking dental professional. Subsequently, it underwent translation and scrutiny by team members proficient in the Greek language. Ten orthodontists, including faculty members and postgraduate students, voluntarily participated in interview-based sessions to assess comprehensibility. The questions for the study were derived from a comprehensive review of the literature on the topic as previously described. At the outset of the process, we identified 15 primary questions, drawn from the literature review, which served as the foundation for the final questionnaire employed in the study.

The final questionnaire had a first part describing the instructions for participation (Appendix A) and four distinct sections with questions (Appendix B). Part 1 consisted of 10 questions regarding the participants, which included gender, age, country, marital status, children, degree in dentistry, other degrees, years in specialty, employment status, and team members. Part 2 gauged the perceived importance of 15 identified risks from the initial survey round, featuring questions Q1–Q25. These risks included root resorption; temporary undesired changes to the occlusion; the possibility of sleep difficulties; not achieving an ideal result; the development of black triangles between teeth; the possible need to take additional X-rays; possible speech difficulties; having to use a protective splint during sports activity; the duration of orthodontic treatment and the number of individual visits involved; the possibility of transmission of infectious diseases within the dental office; and swallowing detached brackets or other orthodontic appliances. Respondents utilized a 5-point Likert scale (1 = never, 2 = rarely, 3 = sometimes, 4 = often, 5 = always) to express their evaluations. 

Part 3 focused on practical professional aspects of the orthodontic procedure, incorporating six additional questions (Q26–Q33). These questions addressed professional and practical dimensions, such as cost, marketing, and clinic management; time estimation; timing of the communication approach; and economic impact assessment. Lastly, Part 4 was comprised of two open-ended questions allowing participants to articulate challenges encountered in risk communication and offer suggestions for educational enhancements. The total number of questions in the present questionnaire was 35.

The questionnaire underwent translation and validation in both the Greek and Slovak languages by team members and five professional (non-university) orthodontists in each country. Criteria for participant inclusion were orthodontists and postgraduate orthodontic students, while dental students, general dentists, and orthodontists practicing abroad without specialization were excluded. An e-questionnaire, which was uploaded via Google Forms, and a country-specific QR code were provided for ease of completion. Participants were assured of anonymity, with no collection of personal data in either country. Participation was voluntary, and no incentives were offered. Each orthodontist responded only once, and the questionnaire remained open for three months. Biweekly participation reminders, along with instructions and the questionnaire link, were disseminated by the secretariat of each association. Authorization to send the link to participants in Slovakia was given by the Slovak Orthodontic Society (committee decision on 22 September 2022) which has a total number of 210 members (NS1). In Greece, there are four main orthodontic societies. The Greek team obtained approval from the Greek Association for Orthodontic Study & Research (committee decision on 13 September 2022) for sending the questionnaire to their 493 members (NS2) (total number of orthodontists in Greece: 575). In both countries, all procedures were performed in compliance with relevant laws and institutional guidelines and were approved by the appropriate authorities. Informed consent was obtained from participants at the time they submitted the questionnaire.

### 2.2. Statistical Analysis

Descriptive statistics were performed for all the parts of the survey. For inferential analysis, the outcome variables were the scores of Part 2 (Q1–Q27) survey questions, ranging from 1 (‘never’) to 5 (‘always’). The data were approximately normally distributed and after log transformation, the data followed a normal distribution, according to the Shapiro–Wilk test for normality (*p* > 0.05). Therefore, potential bivariate associations between orthodontists’ socio-demographic characteristics and the scores for Part 2 survey questions were assessed using *t*-tests and ANOVA. In addition, multiple linear regression models were applied, having as predictors the demographic characteristics, and as outcomes the scores for Part 2 survey questions. All reported probability values (*p*-values) were compared with a significance level of 5% (*p* < 0.05). The analysis of coded data was carried out using IBM SPSS Statistics for Windows, Version 28.0. Armonk, NY, USA: IBM Corp.

## 3. Results

N1 (91) questionnaires were filled in Slovakia (response rate = 43.3%) and N2 (77) questionnaires were filled in Greece (response rate = 15.61%). The internal consistency of Part 2 of the survey (Q1–Q27) was very satisfactory (Cronbach’s alpha: 0.87).

Demographic characteristics showed that the sample consisted of 115 females (68.5%) and 53 males (31.55%). Most participants (25.0%) were between the ages of 41 and 50, with the lowest percentage (11.19%) being under the age of 31. Furthermore, most respondents (67.9%) were married with one or two children (54.2%). In terms of academic qualifications, 68.4% had a master’s degree, while 17.3% had a PhD. Most of the participants had 1–10 years of experience in the field of orthodontics, while 28% of them had been practicing orthodontics for 11–20 years. 

Statistically significant differences were found in the frequencies of higher education between the two countries. Greek orthodontists reported master’s and PhD degrees in significantly higher frequency compared to their Slovak counterparts (72/77 = 93.5% vs. 72/91 = 79.1%, respectively, χ^2^ = 16.832, *p* < 0.001) as shown in Table 1.

In Part 2, we indicated that some risk assessment questions received a higher score of importance from participants. As seen in Figure 1, these questions were (Q5) “Do you mention the possibility of tooth root resorption?”, (Q6) “Do you mention the possibility of tooth necrosis?”, (Q13) “Do you mention the possibility of temporary undesired changes to the occlusion?”, (Q18) “Do you mention the possibility of sleep difficulties?”, and (Q20) “Do you mention the possibility of not achieving an ideal result?”.

The results from bivariate associations between the risk/events communication survey question scores and orthodontists’ socio-demographic parameters are shown in Table 2.

A country-wise comparison revealed statistically significant differences in responses to questions Q1, Q5, Q7–10, Q13, Q15–18, Q20–22, and Q27. Specifically, Greek orthodontists expressed higher concern about potential risks, such as relapse of orthodontic treatment; root resorption; temporal occlusal changes; and failure of desired movement of specific teeth. They also showed greater apprehension about the following: extractions during treatment; additional X-rays; an idiopathic inability of tooth eruption; inclusion or ankylosis of teeth; failure to achieve the desired outcome; protective splint during sports activities; and difficulties in mastication, speech, and sleep. Moreover, significant differences emerged in the communication of the duration of orthodontic treatment and the possibility of emergency visits due to practical problems. Slovak practitioners tended to be interested more in the following, in diminishing order: sleeping difficulties; temporary undesired changes in occlusion; not achieving an ideal result; using a protective splint during sports activities; duration of treatment; and number of individual visits. They also tended to obtain written or digital consent from patients or their parents/guardians more frequently than the Greek team.

In terms of gender comparison, significant differences were found in responses to questions Q1, Q6, Q7, Q10, Q12, Q19, and Q27. Male participants appeared more inclined to discuss the risks of relapse; the failure of desired movement of some teeth; the need for extractions during treatment; possibly finding an idiopathic inability to erupt; inclusion or ankylosis of teeth; tooth necrosis; and the need for modified oral hygiene instructions when compared to their female counterparts. Conversely, females obtained written or digital consent from their patients or their patients’ parents/guardians more frequently.

Regarding degree comparison, statistically significant differences were observed only in questions Q1 and Q12. PhD respondents were more likely to emphasize the risk of failure of the desired movement of some teeth and the possibility of finding an idiopathic inability to erupt, inclusion, or ankylosis of teeth.

Experience in the profession demonstrated a statistically significant difference in response to question Q6. Orthodontists with 1–10 years and those with 31 or more years of experience communicated the risk of tooth necrosis more frequently than their counterparts, and notably, at the same frequency.

The results of multiple linear regression analysis in Table 3 indicated that Slovak orthodontists reported significantly lower scores for survey questions Q1, Q7, and Q27 of Part 2 compared to their Greek counterparts, after adjusting for all their socio-demographic characteristics (gender, age, marital status, highest degree in dentistry, years in profession, number of children). This suggests that socio-demographic factors may contribute to variations in the perception and communication of risks/events among orthodontists from different countries.

In questions Q28–Q33 we collected data about the marketing habits and procedures of the participants, shown in Table 4. According to marketing habits, 89.3% of participants outline the risks again during orthodontic treatment. Additionally, 86.3% of doctors communicate the risks to their patients themselves. The average time spent on patient information is 15 to 30 min, with the majority using digital images or videos. Moreover, it appears that the clinic’s website is the most frequently used marketing communication tool.

In Part 4, participants’ views on frequent challenges during orthodontic risk communication and assistive tools to resolve the issues are reported. It appears that the lack of patient interest/cooperation, lack of time, patient misinformation (low health literacy, and the difficulty of patients remembering instructions are the most common problems faced by professionals when communicating risks (Table 5).

To further address our data collection from Part 4 of the questionnaire, participants’ original views on the ethical consensus on orthodontic risk perspectives proposed that professionals should use more time to address the procedure. Also, written information should be given to patients according to the standards for informed consent of the American Association of Orthodontists (AAO), which ideally could be sent to the interested parties by email prior to their clinical appointment so that they have time to read it and discuss any questions that may arise. 

Given all the data collected, the relevant protocol for informed consent, derived from this study, is presented in Table 6.

## 4. Discussion

The communication of therapeutic risks is a cornerstone of health sciences, forming the basis for valid consent; shared decision-making; and the delivery of person-centered care [35]. This study investigated the perspectives of orthodontists in two European countries, shedding light on the issues of risk communication in orthodontic practice. Orthodontists in both countries seem to acknowledge the importance of risk communication and assess risk accordingly but may not consistently communicate certain treatment risks. This was also indicated by Bernabe et al.’s findings, which highlight the omission of risks, for example, related to eating and speaking [36]. This finding echoes the common clinical dilemma of deciding which risks should be communicated, a decision that is influenced by various factors, such as the orthodontist’s social characteristics, gender, and experience, as well as demographics, as mentioned elsewhere [32]. 

More specifically, in our study, a comprehensive country-wise comparison exposed notable distinctions in the types of risks prioritized and the modalities of consent employed by orthodontists. Greek practitioners exhibited a distinct focus on communicating potential risks, emphasizing aspects such as relapse; root resorption; temporal occlusal changes; and failure of desired movement. This emphasis aligns with the findings of studies, such as those by Cohen and Yen (2014) [37] and Hancox et al. (2014) [38], which have discussed the significance of informed consent in orthodontic treatment and the variations in emphasis on these specific risks. Conversely, Slovak orthodontists displayed a focus on sleeping disorders; tooth resorption; temporary occlusion changes; not achieving an ideal result; and a preference for written or digital consent practices, reflecting a divergence in communication strategies that corresponds with insights from studies such as those by Kumar et al. (2016) [39] and Greco (2023) [34]. The contextual variations highlighted in these studies underscore the influence of regional factors and professional practices on the theme of risk communication in orthodontics, as further expounded elsewhere [40]. 

A closer examination of gender and degree comparisons in our investigation revealed noteworthy variations in risk communication strategies among orthodontists. Male practitioners demonstrated a higher frequency of discussing specific risks, highlighting factors such as relapse, failure of desired movement, and the potential for extractions during treatment. On the other hand, female orthodontists exhibited a predilection for obtaining written or digital consent, reflecting a distinct approach to the communication of treatment risks. These findings align with insights from other studies [38,39], which report on gender-related differences in orthodontic risk communication practices.

Furthermore, in our investigation, an in-depth analysis of degree comparisons in orthodontic risk communication revealed distinctive patterns. Orthodontists with a PhD degree demonstrated a heightened awareness of specific risks, particularly emphasizing the importance of discussing the failure of the desired movement of certain teeth and the potential discovery of an idiopathic inability to erupt. These findings align with the insights provided by studies like that by Perry et al. (2021) [32], which explore the impact of professional qualifications on risk assessment and communication. The observed variations in risk communication strategies based on academic degrees underscore the necessity for tailored approaches that consider individual characteristics and diverse professional backgrounds. This notion is further supported by other studies [34,40] emphasizing the importance of understanding how educational background influences risk communication practices in orthodontics.

Our outcomes highlighted specific orthodontist concerns, with root resorption, root necrosis, temporary undesired changes to the occlusion, not achieving an ideal result, the duration of orthodontic treatment, and the potential for sleep disorders emerging as the most critical topics for discussion. These findings contribute to the ongoing discourse on orthodontic risk communication, complementing the evidence-based approach advocated by Perry et al. (2021) [32], and offering another perspective on the risks prioritized by orthodontic practitioners in diverse clinical settings. These specific risks should be addressed during the informed consent process, ideally being emphasized through verbal communication by the orthodontist, as revealed by our data. Discrepancies between our study’s results and existing data suggest potential influences of socio-demographic parameters and the COVID-19 pandemic, particularly with issues related to the transmission of infectious diseases in the dental clinic [41,42]. These differences underline the dynamic nature of risk communication in orthodontics, whereas external factors and evolving contexts contribute to variations in practitioners’ perspectives and practices such as the ones addressed in our study.

The practical implementation of informed consent also remains a subject of investigation. Carr’s study [43] suggested a verbal review of consent, focusing on the initial points presented in a slide presentation, while Carter and Al-Diwani (2022) [44] found no significant differences among different methods of informed consent. Skulski et al.’s study [45], on the other hand, emphasized the positive impact of rehearsal interventions on recall and comprehension, which highlighted the potential benefits of incorporating educational strategies into the consent process [43,44,45]. In our study, orthodontists exhibited a modernized approach to patient communication by frequently employing digital images and videos for the presentation of treatment risks, as also discussed by Lee et al. (2006) [46] and Terry and Cain (2016) [47]. This contemporary method underscores the field’s recognition of the importance of visual aids in enhancing patient understanding. The prevalent adoption of obtaining written or digital consent, in line with current technological trends in healthcare, reflects the broader integration of technology into orthodontic practices [48]. This aligns also with the shifting landscape of healthcare towards digitalization and emphasizes the orthodontic community’s commitment to staying abreast of technological advancements. Additionally, the reported repetition of risk discussions during orthodontic treatment in our data is in accordance with the findings of Kellar (2009) [49], where a written consent practice was observed in 89.3% of participants. 

Moreover, the incorporation of risk assessment into orthodontic practice is a multifaceted endeavor, as derived from our findings, transcending clinical realms to involve strategic professional management and marketing considerations, as also discussed elsewhere [50,51]. This holistic approach aligns with the ethical imperative of transparent communication, where potential risks are explicitly conveyed to patients before treatment initiation [52]. Such communication is not merely a moral obligation but a pivotal aspect contributing to patient awareness and preparedness for any conceivable adverse conditions that might arise [53]. Our findings emphasize the significance of shared decision-making and patient autonomy in the clinical encounter, acknowledging that informed patients are better equipped to actively participate in their orthodontic journey. This aligns with the evolving landscape of healthcare, where transparency not only serves ethical principles but is also instrumental in fostering a patient-centered approach and optimizing overall treatment outcomes [34].

The study’s limitations include an exclusive focus on orthodontists in two European countries, potentially limiting global generalizability. The significant difference in response rates between Slovakia (43.3%) and Greece (15.61%) raises also concerns, emphasizing the need for more tailored approaches [54,55]. But our findings align with global trends, suggesting that healthcare professionals often exhibit lower response rates than patients [56]. The challenges in surveying specific professional groups, such as healthcare professionals, require specialized approaches, as mentioned elsewhere [57]. Future research should then consider diverse survey administration methods [58]. Additionally, this study notes the influence of socio-demographic parameters on risk communication but calls for a more in-depth exploration of factors like age, gender, and professional experience. The impact of the COVID-19 pandemic on risk communication and effective strategies for presenting informed consent information also warrant future investigation. Long-term studies assessing the impact of effective risk communication on patient satisfaction and treatment outcomes would contribute to best practices in orthodontic care. These considerations show the complexity of factors influencing survey response rates, emphasizing the need for tailored and informed approaches in future research initiatives.

Despite its limitations, this study, focusing on the incorporation of a communication protocol into orthodontic practices in Greece and Slovakia, carries substantial benefits not only for the two mentioned countries but also for the global orthodontic community. By enhancing the informed consent process, practitioners are poised to elevate patient care, promoting a model that aligns with contemporary healthcare trends emphasizing ethics, transparency, and patient-centered care. The tailored communication protocol considers individual and cultural variations, addressing the specific needs and preferences of patients in diverse contexts. This approach not only respects the unique characteristics of patients but also contributes to the globalization of best practices in orthodontics. The findings emphasize that a one-size-fits-all model may not suffice in meeting the diverse needs of orthodontic patients worldwide. This research thus advocates for a paradigm shift toward a more personalized and culturally sensitive communication approach, fostering the highest standards of care and ensuring global patient satisfaction in orthodontic practices.

## 5. Conclusions

In this descriptive study, significant differences were found in the orthodontic risk perspectives of orthodontists during orthodontic treatment between Greece and Slovakia. Orthodontists in both countries seem to acknowledge the importance of informed consent about orthodontic risk perspectives and assess risk accordingly, but they may not consistently communicate certain treatment risks. The variations in risk communication practices, influenced by socio-demographic factors, underscore the need for standardized guidelines. 

## Figures and Tables

**Figure 1 healthcare-12-00492-f001:**
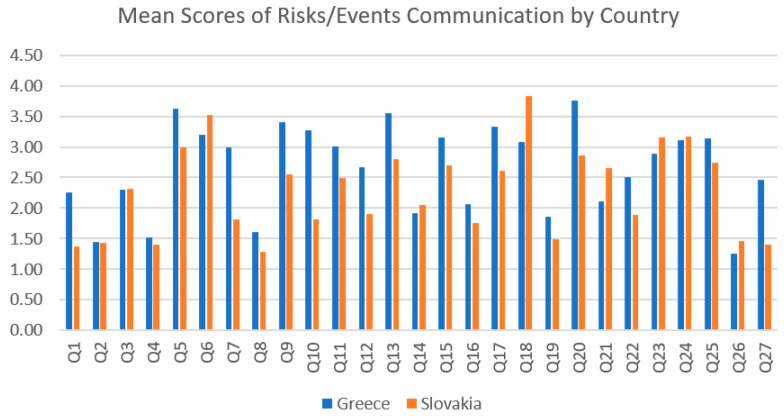
Mean scores of risk assessment questions for Greece and Slovakia. Note: The horizontal axis represents the mean score of risk/events communication; Q: Question of the survey in Appendix B.

**Table 1 healthcare-12-00492-t001:** Frequency of higher education among participants.

	What is Your Highest Degree in Dentistry?						
	Bachelor		Master		PhD		Total
	N	%	N	%	N	%	N
Greece	5	20.8%	50	43.5%	22	75.9%	77
Slovakia	19	79.2%	65	56.5%	7	24.1%	91
Total	24	100.0%	115	100.0%	29	100.0%	168

**Table 2 healthcare-12-00492-t002:** Bivariate associations between risk/events communication survey question scores and orthodontists’ socio-demographic parameters. Only significant associations are depicted.

	Country				Gender				Highest Degree in Dentistry *						Years in Profession as Orthodontist **							
Survey Item	Greece		Slovakia		Males		Females		Bachelor		Master		PhD		1–10		11–20		21–30		>30	
	M	SD	M	SD	M	SD	M	SD	M	SD	M	SD	M	SD	M	SD	M	SD	M	SD	M	SD
Q1	2.26	1.25	1.37	0.49	2.15	1.2	1.61	0.88														
Q5	3.62	1.25	3.00	1.23																		
Q6					3.11	1.24	3.50	1.05							3.50	1.02	3.49	1.02	2.84	1.16	3.50	1.25
Q7	2.99	1.15	1.81	0.77	2.62	1.18	2.23	1.09	2.08	0.93	2.29	1.12	2.83	1.19								
Q8	1.61	1.00	1.29	0.68																		
Q9	3.40	1.41	2.55	1.24																		
Q10	3.27	1.73	1.81	1.08	2.91	1.71	2.29	1.5														
Q11	3.01	1.60	2.49	1.28																		
Q12	2.68	1.23	1.90	0.98	2.51	1.3	2.14	1.08	1.88	0.95	2.17	1.12	2.26	1.16								
Q13	3.56	1.29	2.80	1.38																		
Q15	3.16	1.59	2.70	1.33																		
Q16	2.06	1.13	1.76	0.84																		
Q17	3.34	1.59	2.60	1.41																		
Q18	3.08	1.19	3.84	1.09																		
Q19					2.09	1.49	1.46	0.96														
Q20	3.75	1.43	2.86	1.19																		
Q21	2.10	1.51	2.66	1.34																		
Q22	2.51	1.68	1.89	1.09																		
Q27	2.47	1.50	1.41	0.83	2.21	1.52	1.75	1.15														

Note 1: Higher scores indicate a greater frequency of risks/events communication. Note 2: For the country and gender, *t*-tests were applied, for the highest degrees and years in the profession, ANOVA was used. Note 3: M = mean score, SD = standard deviation. Note 4: Grey cells reflect non-significant results; therefore, the relevant means and standard deviations were not included to avoid misinterpretation of the data. * Post hoc pairwise analysis: the PhD group’s scores were significantly higher than those of all other groups. ** Post hoc pairwise analysis: the scores of the 21–31 years age group were significantly lower than those of all other groups.

**Table 3 healthcare-12-00492-t003:** Multiple linear regression analysis between risk/events communication survey question scores and socio-demographic characteristics in orthodontists (only outcomes with significant associations are depicted).

Outcome	Predictor	*B*	95% CI	*Beta*	*T*	*p*
Q1 score	Country *	−0.81	−1.11, −0.51	−0.40	−5.34	0.001
	Gender	−0.24	−0.56, 0.08	−0.11	−1.46	0.147
Q7 score	Country *	−1.17	−1.50, −0.84	−0.52	−7.10	0.001
	Gender	0.07	−0.28, 0.41	0.03	0.39	0.695
	Highest Degree in Dentistry	0.07	−0.21, 0.35	0.04	0.51	0.615
Q27 score	Country *	−0.99	−1.39, −0.60	−0.38	−5.03	0.001
	Gender	−0.09	−0.51, 0.32	−0.04	−0.46	0.644
	Number of Children	−0.13	−0.30, 0.04	−0.11	−1.53	0.128

* Slovakia vs. Greece. Note: Gender, age, marital status, highest degree in dentistry, years in the profession, number of children, and country, in different combinations, were included in multiple linear regression models and the models with the best fit were selected to report significant coefficients.

**Table 4 healthcare-12-00492-t004:** Practices of communication and marketing.

Practices of Communication and Marketing	Score
Frequency of recurrence of the risk communication process during the progress of orthodontic treatment	89.3%
Frequency of orthodontists being the main source of communication	86.3%
Mean time spent informing patients in the first appointment	15–30 min
Common communication tools	Digital images or videos
Most frequent marketing communication tool	Clinic’s website

**Table 5 healthcare-12-00492-t005:** Frequent challenges when communicating risks.

Frequent Challenges when Communicating Risks	% Percentage
Lack of patient cooperation/communication	37.5
Lack of time	16.1
Low health literacy/misinformation (internet, general dentists, etc.)	8.3
No problem	27.4

**Table 6 healthcare-12-00492-t006:** Risks and proposed actions for a tailored communication approach for orthodontic patients.

Risks	Proposed Actions
Comprehensive risk discussion	Ensure a detailed and comprehensive discussion of potential risks associated with orthodontic treatment. Emphasize risks such as root resorption, temporary changes to occlusion, sleep difficulties, failure to achieve ideal results, and other critical factors identified in the study.
Visual aid utilization	Incorporate visual aids such as digital images and videos during the informed consent process. This aligns with contemporary trends and enhances patient understanding of potential risks and treatment procedures.
Written or digital consent	Provide options for written or digital consent. Acknowledge the prevalence of digital trends in healthcare and allow patients to choose their preferred mode of providing consent.
Repetition of risk discussions	Encourage orthodontists to repeat risk discussions during treatment. This repetition enhances patient comprehension and awareness throughout the orthodontic journey.
Tailored approaches based on individual characteristics	Recognize and adapt the consent process based on individual characteristics, such as gender and professional background. Acknowledge that male orthodontists may prefer discussing specific risks more frequently, while female orthodontists may lean towards obtaining written or digital consent.
Cultural sensitivity	Consider cultural differences in risk communication.
Continued education	Promote continued education for orthodontists on effective communication strategies and informed consent. This ensures that professionals stay updated on best practices and contribute to ongoing improvements in patient care

## Data Availability

Data are available upon request.

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
