# Peer review of "Orthodontic Risk Perspectives among Orthodontists during Treatment: A Descriptive Pilot Study in Greece and Slovakia"

_healthcare, 2024, doi:10.3390/healthcare12040492_

Round 1
Reviewer 1 Report
Comments and Suggestions for Authors
A Professional Consensus on Orthodontic Risks: Communication Approach, Quality Assurance, and Educational Strategies
Reviewer Report
Thanks to the authors for the study. Authors conducted a study investigating different approaches to risk perception in orthodontics. The study results revealed differences in terms of perspective on the risks encountered in orthodontic treatments according to regional, gender and education level. In this sense, it was seen that there is no common consensus regarding the risks in orthodontics, but approaches differ depending on the orthodontist’s individual experience, education level, etc. It is important to uncover these results. However, there are parts of the study that need to be developed and improved before publication.
Here are my opinions on the study requiring a major revision:
ABSTRACT:
Line 24: ‘resorption’ should be used instead of ‘necrosis’
General comment: The abstract is well written, covering important aspects of the study.
INTRODUCTION:
General comment: The introduction is a little short. Since a lot of scientific data can be easily accessed in terms of the questions researched in the study, the introduction section should be expanded a little further by supporting it with current literature. The importance of survey studies can also be mentioned. Additionally, in a separate paragraph, the deficiencies in the literature on the subject under investigation and the contribution of the study should be mentioned.
Line 81-82: It would be appropriate to examine current studies that support these statements. ‘This situation has been reported in current studies using different types of orthodontic brackets or adhesive agents, that inadequate oral hygiene causes demineralization.’ For this current statement, it is recommended to check these studies: https://doi.org/10.3390/ma16030984 and https://doi.org/10.3390/coatings13020401
Line 87-89: Attention should also be drawn here to the additional time and cost required to re-bonding orthodontic brackets after they detached, and to the demineralization caused by re-bonding. For this reason, this current study can be checked: https://doi.org/10.29058/mjwbs.1265876
Line 92-97: It is necessary to state more clearly the gap in the literature on which the study was planned and the purpose. It’s a bit complicated. It should be revised.
Line 97-98: ‘This protocol is envisioned as an integral component of the consent process, contributing to the delivery of high-quality orthodontic treatment.’ Is this statement the hypothesis of the study? It should be revised accordingly.
MATERIALS AND METHODS:
General comment: In the materials and methods section, the procedures applied during the study should be explained in all ethical and technical aspects. It is appropriate to indicate and cite the articles used when forming research questions for this study. However, it is not appropriate to write this as a discussion section. Therefore, a comprehensive revision of this section in a simple, elegant and understandable manner is required.
Line 101-136: These statements are not related to the materials and methods section. With an appropriate revision, it should either be added to the introduction or discussed in the discussion section. Statements regarding the purpose of the study are stated at the end of the introduction and do not need to be stated here. Again, the background regarding the research topic should be explained in the introduction.
RESULTS:
These research questions should also be presented as a table in the material method, except for the appendix.
Line 211-216: These statements should be stated in the materials and methods section.
Line 217-218: What the scores and abbreviations on the horizontal and vertical axis mean should be stated below the Figure 1.
Line 223: Table 3 is difficult to understand. To be more understandable, p values should be revised by giving them separately in the table.
Line 224: On what basis was the distinction between high and low scores applied? Which score and after should high/low be stated? And this should be stated in the materials and methods section.
DISCUSSION:
Line 395-400: No tables are presented in the discussion section. It should be deleted from here. It should be added to the Results. Here, the data in the table should be discussed in relation to the literature.
Line 421-482: It is sufficient to add a paragraph containing the strengths and limitations of the study. It is not appropriate to discuss these at length in a separate section. A few simple, understandable paragraphs are sufficient. Therefore, what is written under the title ‘Limitations of the study’ should be combined with the discussion section and revised.
CONCLUSION:
It is written in a sufficiently enlightening manner.
REFERENCES:
References from old years need to be replaced with current literature. And some references need to be revised according to journal rules.
Comments on the Quality of English LanguageModerate editing of English language required.
Author Response
Thank you for your comments. We provide a revised manuscript according to suggestions. Please see yellow underline text.
All the best
The authors

Reviewer 2 Report
Comments and Suggestions for Authors
Dear Author,
Please find the comments attached
Regards

Author Response
Thank you for your comments. We provide a revised manuscript according to suggestions. Please see yellow underlined text.
All the best
The authors

Reviewer 3 Report
Comments and Suggestions for Authors
I have reviewed the manuscript entitled: "A Professional Consensus on Orthodontic Risks: Communication Approach, Quality Assurance, and Educational Strategies".
I, as a reviewer, consider this manuscript to be unfit for publication. The authors, I believe, should undertake a major analysis and modification of the submitted work.
- Does the title of the manuscript clearly describe the aim of the study?
- The length of the abstract, introduction and discussion is excessive. The
- The number of keywords is excessive.
- Is it essential for the introduction to have 31 bibliographical references?
- Do the authors consider a total number of references of 90 to be reasonable?
- How was the sample size calculated? The sample size was very small.
- In my humble opinion, I think the authors should review the citation of such a large number of self-published articles.
Author Response
Thank you for your comments. We provide a revised manuscript according to suggestions. Please see yellow underline text in the revised draft.
All the best
The authors

Round 2
Reviewer 1 Report
Comments and Suggestions for Authors
The manuscript has improved significantly. All points have been addressed.
Comments on the Quality of English LanguageMinor editing of English language required
Author Response
Further changes underlined with green are to be seen across the manuscript.
Thank you for your time and effort to review our work
the authors

Reviewer 3 Report
Comments and Suggestions for Authors
The authors have not made a major modification to their manuscript to improve it.
Author Response
We have further processed the manuscript. Please see changes in green underline./ Thank you for your time
The authors
